# Risk Factors for Relapse in People with Severe Mental Disorders during the COVID-19 Pandemic: A Multicenter Retrospective Study

**DOI:** 10.3390/healthcare10010064

**Published:** 2021-12-30

**Authors:** Antonio José Sánchez-Guarnido, Paloma Huertas, Rosario Garcia-Solier, Miguel Solano, Beatriz Díez, Marta León, Javier Herruzo-Cabrera

**Affiliations:** 1Santa Ana Hospital, 18009 Granada, Spain; antonioj.sanchez.guarnido.sspa@juntadeandalucia.es (A.J.S.-G.); martaleonruiz@hotmail.com (M.L.); 2Axarquía Hospital, 29700 Málaga, Spain; 3Virgen de Valme University Hospital, 41014 Sevilla, Spain; rosario.garcia.solier.sspa@juntadeandalucia.es; 4Infanta Leonor University Hospital, 28031 Madrid, Spain; miguel.solano@salud.madrid.org; 5José Germain University Hospital, 28911 Madrid, Spain; beatrizdiezcto@gmail.com; 6Department of Psychology, Faculty of Education Sciences, University of Córdoba, 14071 Córdoba, Spain; ed1hecaf@uco.es

**Keywords:** COVID-19, severe mental disorders, associated factors

## Abstract

Background: Evidence suggests that different variables associated with the COVID-19 pandemic may increase the risk of relapse in people with Severe Mental Disorders (SMDs). However, no studies have yet looked closely at the different risk factors involved to determine their influence on the worsening of these patients’ illnesses. Objective: To analyze which variables related to the COVID-19 pandemic have increased the risk of relapse in patients with SMDs. Method: A multicenter retrospective cohort study in which data were collected from 270 patients with mental disorders who had been under follow-up in day hospitals during the year 2020. Results: The proportion of full mental health inpatient admissions was significantly higher in those who lost their employment (40.7% vs. 18.1%; *p* = 0.01), in those who were not receiving psychotherapy interventions (33.9% vs. 16.6%; *p* = 0.006), and in those who were not receiving occupational therapy (25.7% vs. 13.6%: *p* = 0.013). Significant associations were detected between urgent mental health consultations, the number of COVID-19 symptoms (B = 0.274; *p* = 0.02), and the low-income group (1.2424 vs. 0.4583; *p* = 0.018). Conclusions: COVID-19 symptoms and certain consequences of the pandemic, such as loss of employment, economic hardship, and loss of interventions, have brought about clinical worsening in people with SMDs. Knowledge of these factors is important for health-related decision-making in future outbreaks or pandemics.

## 1. Introduction

According to the United Nations [1], although the new coronavirus pandemic (COVID-19) is primarily a physical health crisis, it may also be the source of a major mental health crisis. It is very likely that there will be an increase in and worsening of mental disorders secondary to this crisis. A recent meta-analysis indicated that the prevalence of depression, anxiety, and post-traumatic stress in populations affected by COVID-19 is more than three to five times higher than in the general population [2]. In relation to the psychotic disorder spectrum, Hu et al. [3] found a positive relationship between the incidence of schizophrenia and the epidemic situation in China. In the same vein, another retrospective case registry study reported a rate of 0.9% of first psychotic disorder diagnosis or relapse after COVID-19 diagnosis, substantially higher than for all other events studied [4].

Different factors can increase mental health problems in the population. In this regard, social distancing, a measure used in several countries to attempt to decrease community transmission of COVID-19, can have particularly negative effects [5]. In a community survey in China, fears of contracting COVID-19, coupled with social isolation, led to more than half of the population (53.8%) showing moderate or severe psychological impacts, with 16.5% developing moderate to severe depressive symptoms, 28.8% moderate to severe anxiety symptoms, and 8.1% moderate to severe stress levels [6]. In Spain, a study with a sample of 1161 adults found high levels of sleep problems and emotional symptoms (worry, stress, hopelessness, depression, anxiety, nervousness, and restlessness) during lockdown [7]. Other studies have found similar results and have also identified some risk factors associated with increased mental health problems, such as having family members or acquaintances infected with the virus [8], abuse of social networks [9], and an excess of information through the media [10]. Conversely, as protective factors, it appears that living in urban areas, living with parents, and belonging to a family with some degree of financial stability may prevent mental health problems [8].

While the COVID-19 pandemic, its associated stress problems, and the adoption of public health measures such as lockdowns worsen mental health in the general population, the impact of the virus and its treatment is expected to be even greater in people with previous severe mental disorders (SMDs) due to their greater susceptibility to stress compared to the general population [11,12,13]. The pandemic may have a broader negative impact on the mental health status of people living with SMDs, creating an adverse environment for members of vulnerable groups, such as those with prodromal symptoms and patients with schizophrenia. A recent study carried out in Spain found that exposure to all of these circumstances led to a greater increase in anxiety in the SMD population than in the population with no psychopathological diagnoses [14]. In this case, COVID-19-related public health measures, such as lockdowns and quarantines for suspected cases, together with the socio-economic repercussions of such measures, appear to negatively affect the mental health status of people with SMDs. The COVID-19 pandemic may therefore act as a catalyst for the re-emergence of psychosis [11] or for the symptomatological exacerbation of individuals with SMDs [15,16]. Another aspect that may be affected is the ability of people with SMDs to have their basic needs met, given the fact that many of these patients rely on certain community support services (employment, occupational, mutual support, etc.) that may be difficult to access in these circumstances [17]. Since the start of the pandemic, there have been reports of disrupted drug supply chains [18,19] and the suspension of community services on which many people with SMDs rely [20,21]. The change in environment associated with COVID-19, the disruption of mental health and community services, and the consequent reduction in social interaction, may therefore all have a particularly negative effect on people with SMDs [22]. Recovery from such illness is associated with social support [23], and we know that extensive community support, including occasional contacts in pharmacies, supermarkets, shops, and cafés, improves community integration for SMD patients [24]. On the other hand, specific restrictions affecting mental health follow-up are likely to lead to symptomatological worsening and higher rates of relapse in this population.

COVID-19 infection itself or the suspicion of infection may function as a risk factor for symptomatological destabilization and/or increased relapse rates in hospitalized SMD patients suspected of having contracted COVID-19. This was first suggested by Liu et al. [25] in a study comparing two groups of mental health inpatients, one group with suspected COVID-19 and one group with no suspicion of infection. The authors observed an increase in stress and depressive and anxiety symptoms in the group with suspected COVID-19, as well as poorer sleep quality. Along the same lines, González-Blanco et al. [14] report significantly higher levels of anxiety in patients with SMDs suffering from coronavirus than in the general population. Moreover, the “infodemic” brought about by COVID-19-related information overload may exacerbate the clinical delusions, hallucinations, and disorganized thinking commonly experienced by individuals with SMDs [15], as well as impact the nature and content of psychotic pathologies in individuals with or at risk of developing psychosis [11,26]. There is evidence that under pandemic circumstances, patients with psychosis prefer to accept psychotic explanations [27].

Beyond COVID-19 infection itself or the suspicion of it, some studies report a positive association between high levels of anxiety or even post-traumatic stress disorder and having a family member infected with the virus [28,29]. Family members who have lost a loved one during the pandemic and were separated from ill loved ones may also be vulnerable to psychiatric illness [30]. In a previous study, it was estimated that 50% of the family members of SARS-CoV-1 patients experienced psychological problems (mainly depressive symptoms) and stigmatization [31]. Such a life event is likely to have a greater impact on people with previous severe mental pathologies, and failure to address the urgent needs of those experiencing loss and bereavement may result in the worsening of mental and physical health problems [32].

Other psychosocial circumstances, such as unemployment and economic problems, which have increased significantly during the COVID-19 pandemic, have been shown to be risk factors for mental health [30,31,32,33] and, more specifically, for the development of SMDs and future relapse [34,35,36,37].

Evidence from the literature suggests that different variables associated with the pandemic may increase the risk of relapse in people with SMDs. Knowledge of these factors could aid health care decision-making in future outbreaks or pandemics. Our study therefore aims to analyze which variables related to COVID-19 (COVID symptoms, bereavement, loss of employment, loss of interventions) increase the risk of relapse in patients with SMDs. Since even under normal conditions these variables in themselves increase the risk of full mental health inpatient admissions and urgent mental health consultations, we hypothesize that in pandemic conditions these risks will have increased.

## 2. Materials and Methods

A sample size of 272 patients was calculated for a confidence level of 95%, a statistical power of 80%, and an estimated relapse rate of 30% in the control group and 20% in the intervention group, with possible losses of 15%. Using sampling stratified by region to facilitate generalization to the Spanish population, 15 mental health day hospitals (MHDHs) of the Spanish National Health System took part in the study. All patients from these MHDHs who met the criteria established in the participants section were included in the study. 

The inclusion criteria were people over 18 years of age of both sexes, diagnosed with mental disorders and who had been under follow-up in MHDHs during 2020. The only exclusion criteria were a main diagnosis of mental retardation and the refusal to participate voluntarily in the study. The study was based on a definition of SMDs not only at the level of psychopathological diagnosis but also in terms of the intensity of the intervention required. This criterion was operationalized by selecting patients who had required follow-up during this period in Spanish National Health System MHDHs.

In the end, 15 MHDHs participated. The sample consisted of 270 patients, made up of 120 men and 150 women, aged between 18 and 67 years and with an average age of 39.9. The most frequent diagnoses were severe disorders such as schizophrenia or other psychotic disorders (30.4%), personality disorder (27.8%), and bipolar disorder (10.4%). Most of the sample had primary (35.8%) or secondary (41.5%) levels of education, with 14.8% having received a university education. Most of them lived with their family of origin (42.8%), in their own family home (29%), or alone (16.7%). 29.3% of the patients were retired, 26.3% were unemployed, 20% were temporarily unable to work, and 16.7% were employed.

Data were collected retrospectively during the months of October and November 2020 by the staff of each MHDH from medical records and an interview with each patient.

Informed consent was sought from all participants. A database was designed, accessible only to the researchers, and clinical data were processed without patient identifiers.

The following were used as independent variables: the presence of COVID-19 symptoms (fever, chills, tiredness, sore throat, coughing, shortness of breath, headache, nausea/vomiting/diarrhea, anosmia), assessed as the number of symptoms experienced by each patient (during the first wave, PCR testing was not performed on most people with symptoms, so these were our only indicators for the disease); diagnosis of COVID-19 in a family member; death of a family member (first, second, or third degree) due to COVID-19, assessed dichotomously; economic income, assessed dichotomously (above or below 800 euros, which is the average minimum interprofessional wage in the last 4 years in Spain); loss of employment during the pandemic; and lack of psychotherapy, nursing, and occupational therapy interventions.

The socio-demographic variables studied were age, sex, composition of the household where the patient lived, employment status, and level of education.

Two measures of worsening were used as dependent variables: full mental health inpatient admissions within a period of 6 months from the start of lockdown and the number of urgent consultations in mental health services during the same time interval.

Statistical analysis of the data was carried out using IBM SPSS V.21.0 program. (IBM Corp., Armonk, NY, USA) [38].

Percentages were used for the study of categorical variables and mean and standard deviation for the description of quantitative variables.

Chi square was used to analyze the link between dependent variables (full mental health inpatient admission), clinical variables (diagnosis, grief, cases in the setting, loss of intervention, and COVID-19 symptoms), and sociodemographic variables (gender, household composition, level of education, economic income, and loss of employment). To analyze the link with the dependent variable of urgent mental health consultations, Student’s *t* test was used for the dichotomous independent variables (gender, grief, case in the setting, loss of interventions, economic income, and loss of employment) and ANOVA was used for variables with more categories (household composition, diagnosis, level of education). Simple linear regression was used to verify the association between the dependent variable of urgent mental health consultations and quantitative independent variables (age and the number of COVID symptoms presented by the patient).

All analyses were performed with a significance level of α = 0.05.

## 3. Results

### 3.1. Analysis of Full Mental Health Inpatient Admissions

As can be seen in Table 1, the proportion of admissions was significantly associated with patients who had lost their jobs (40.7% vs. 18.1%; *p* = 0.01), those who were not receiving psychotherapy interventions (33.9% vs. 16.6%; *p* = 0.006), and those who were not receiving occupational interventions (25.7% vs. 13.6%: *p* = 0.013). In our sample, the percentage of admissions was also higher in people bereaved by COVID-19, although this association was not statistically significant (30% vs. 19.6%; *p* = 0.258). No significant differences were found for the rest of the potential risk factors (income, environmental factors, nursing intervention, or symptoms compatible with COVID-19), for any of the socio-demographic variables studied (age, sex, household composition, or economic level), or for diagnoses.

### 3.2. Analysis of Urgent Mental Health Consultations

As can be seen in Table 2, there was a significant positive association between the number of COVID-19 symptoms and the number of urgent mental health consultations (B = 0.274; *p* = 0.002). Each symptom of COVID-19 increased the number of urgent mental health consultations by an average of 0.274. The mean number of emergency visits was also significantly higher in people with low incomes (M = 1.24; SD = 4.04 vs. M = 0.46; SD = 1.34; *p* = 0.018; d = 0.26). People in our sample who had suffered the loss of a family member due to COVID-19 presented a greater number of emergencies, although this was not statistically significant (M = 2.35; SD = 5.76 vs. M = 0.92; SD = 5.75; *p* = 0.289; d = 0.30). No significant differences were found for other risk variables (environmental factors, psychotherapy, occupational or nursing interventions, or loss of employment). Neither was there any significant relationship with the socio-demographic variables studied (age, sex, household composition, or economic level) or with diagnosis.

## 4. Discussion

In this multicenter retrospective cohort study based on data collected from 270 patients with SMDs who had been followed up in day hospitals during 2020, different factors related to the COVID-19 pandemic (loss of interventions, COVID-19 symptoms, low economic income, and loss of employment) were found to be associated with an increased risk of relapse.

In particular, full hospitalizations were significantly higher in those who had lost their jobs, in those who were not receiving psychotherapeutic interventions, and in those who were not receiving occupational therapy. In the case of emergency mental health care, there was a significant link between emergency admissions and both the number of COVID-19 symptoms and low income.

The relationships between, on the one hand, a higher proportion of full hospitalizations and loss of employment and, on the other, a greater number of emergency care visits and low income, raises an issue that goes beyond the current social context of the global pandemic. The link between psychosocial factors such as unemployment or economic problems and mental health problems is widely documented [30,31,32,33]. The economic impact of the pandemic is increasingly evident, with unemployment rates, which constitute an established risk factor for mental disorders across a person’s lifespan [30], rising dramatically. Regarding how these factors influence people with SMDs, unemployment appears to increase the likelihood of developing psychosis by 53% [26] and is also a risk factor for relapse in patients with schizophrenia [34,35,37]. Job loss, as a primary loss, also entails losses in areas like economic security, independence, health care, and sense of future [39]. It can therefore be concluded that the socio-economic disadvantages experienced by people with SMDs are particularly likely to put them at risk of suffering the direct and indirect effects of the pandemic [26,40].

The relationship we found between the loss of mental health care and the increase in full hospitalizations concurs with data published by the WHO, which notes in its latest report that the pandemic disrupted critical mental health services in 93% of countries worldwide. This coincided with an increased need for care among specific population groups at particular risk of COVID-related psychological distress, including people with a history of psychopathology [41]. 

In this regard, reduced accessibility to psychotherapy and occupational therapy interventions may have contributed to feelings of abandonment in patients, decreasing their sense of self-efficacy, generating helplessness and uncertainty, and thus increasing the risk of relapse and the need for full hospitalization. The specific restriction of follow-up in mental health can thus lead to an increase in relapses. Different psychosocial approaches have shown efficacy in reducing symptoms, preventing relapse, and maintaining good levels of functioning and well-being in people with SMDs [42,43,44,45,46,47]. Moreover, the dropout rate from psychosocial treatment is markedly lower than the dropout rate from pharmacological treatment [48], which it may also prevent. On the other hand, in a crisis such as the COVID-19 pandemic, emotional distress and a lack of accurate information may have amplified tensions in the social and family environment, and this may have impacted “expressed emotion” [49]. This construct has been widely studied in the literature as a predictor of relapse risk in patients with Serious Mental Disorders [50]. A patient’s loss of psychotherapy interventions may lead to a decline in coping strategies for expressed emotion in the family environment, while their loss of occupational therapy interventions may increase their hours of exposure to that same environment. Furthermore, government-induced restrictions on social activities during lockdown may also lead to secondary losses such as losses of relationships, leisure, and social support [39], and these losses may be intensified by the disruption of occupational interventions, weakening the social support that is associated with recovery. When social contacts are disrupted by social distancing, patient recovery is put at risk [13,24,51,52]. This lack of support constitutes a risk factor for the destabilization of psychotic symptomatology and consequent relapse [53], increasing suicides and exacerbating hallucinatory experiences in individuals with SMDs [23,54,55]. This, too, raises this population’s risk of re-hospitalization.

The statistically significant link between higher numbers of emergency mental health visits and COVID-19 symptoms shown in our study gives us an idea of the fear, worry, anxiety, and general need for emergency hospital care that may be experienced by those experiencing such symptoms due both to their acute discomfort and to the uncertainty regarding the evolution of their illness. Related public health measures such as home isolation may also lead to a worsening of previous psychopathologies. The WHO has already noted an increased need for care among specific population groups at particular risk of COVID-related psychological distress, including people with a history of mental health issues [41]. Furthermore, it may also be possible that COVID-19 drug treatment has a negative influence on the psychiatric evolution of some patients with SMDs. A recent meta-analysis exploring the effect of different coronaviruses on the presentation of mental health symptomatology and the therapeutic approach to the same, documented the relationship between the exacerbation of psychotic or manic symptoms and corticosteroid treatment [56], while a case series study also recently analyzed 10 cases of new diagnoses of psychosis in people with SARS-CoV-2, linking the development of psychosis to both the infection itself and the inflammatory response caused by its treatment [57]. COVID-19 infection and the drugs used to treat it may therefore be associated with psychotic relapses, especially in patients with SMDs.

We believe that one of the main strengths of our study lies in its having focused on people with SMDs, an area where research has to date been scarce despite the fact that this is a particularly vulnerable population. Carrying out a multicenter study allowed us to have a larger sample, with greater potential for generalizing the results. Similarly, the inclusion of different psychopathological diagnoses, although it can be considered a limitation, provided us with a broad overview of what has happened in the SMDs population as a whole. Another advantage, in our opinion, was the use of day hospitals as participating institutions. This provided good records and sufficient time to collect information from patients, resulting in high data reliability.

Nevertheless, the study also has some limitations. An insufficient sample size made statistical significance difficult for variables such as bereavement due to the COVID-related loss of a family member, which in our sample functioned as a risk factor. Additionally, some potential risk factors that could not be studied, such as the degree of social isolation, information overload regarding COVID-19, and the test-based confirmation or exclusion of COVID-19 diagnoses in all patients, are missing. Finally, at the methodological level, conducting a retrospective study always involves a greater risk of bias, although in this regard, the fact that the research was conducted in day hospitals ensured access to good information records and thus reduced that risk. In future research, it would therefore be interesting to incorporate specific diagnoses, larger samples, and additional potential risk factors.

## 5. Future Perspectives and Clinical Implications

The COVID-19 pandemic has increased the risk of worsened mental health among people with SMDs. This worsening has been related to the symptoms of the infection but also to other preventable social circumstances, such as the loss of therapeutic interventions, the loss of employment, and low financial income. Preventive measures therefore need to be developed to increase the well-being of people with SMDs and reduce relapses in future pandemics. These measures should aim to minimize losses in care interventions, introducing alternative non-face-to-face care formats while taking into account the fact that people with SMDs may experience more difficulties in using technology [58,59] and need more support and improved digital skills to access online interventions. The issues of employment and income improvement would also require the development of strong intervention and support programs throughout the process, including vocational guidance, training activities, job searches, preparation for job interviews and for getting and keeping a job, as well as support in coping with potentially stressful situations that may occur in the work environment. For people diagnosed with SMDs and without the possibility of developing an adequate work environment, the system of financial aid for greater autonomy and independence should be more flexible. This would have a decisive impact on this population’s mental health. Further research showing the impact of these factors on mental health is important.

## 6. Conclusions

In summary, our results are in line with WHO predictions that factors such as social isolation, fear of infection, and loss of family members are likely to be compounded by the distress caused by loss of income and, often, employment, increasing mental health problems in the population at the same time that mental health services are becoming less accessible [60]. In addition to the direct consequences of lockdown itself, social distancing measures and the interruption of health and community services, there also exist other factors associated with the pandemic situation (fear of infection by COVID-19, symptoms compatible with the disease, the increase in stressful life events like loss of employment, economic problems, bereavement due to the loss of loved ones, etc.), which may have greater impacts on particularly vulnerable people such as those with SMDs [32,51,61].

## Figures and Tables

**Table 1 healthcare-10-00064-t001:** Bivariate analysis of the proportion of full mental health inpatient admissions as a function of different variables.

Variables	N	Income	χ2 ^a^	*p*
Gender,%(*n*)				
Man	44.4 (120)	16.7 (20)		
Woman	55.6 (150)	23.3 (35)	1.850	0.174
Household composition, %(*n*)				
Family of origin	42.8 (115)	20.9 (24)		
Own family	29 (78)	20.5 (16)		
Single-family	16.7 (45)	19.6 (9)		
Other	11.5 (31)	19.4 (6)	0.053	0.817
Level of education, %(*n*)				
Primary	38.5 (104)	22.1 (23)		
Secondary	41.5 (112)	20.3 (23)		
University students	20 (54)	16.7 (9)	0.673	0.714
Diagnosis,%(*n*)				
Schizophrenia or other psychotic disorder	30.4 (82)	19.5 (16)		
Personality disorder	27.8 (75)	22.7 (17)		
Bipolar disorder	10.4 (28)	32 (9)		
Depressive disorder	9.6 (26)	15.4 (4)		
Other	21.9 (59)	15.3 (9)	3.853	0.426
Economic income, %(*n*)				
More than €800	26.6 (72)	18.1 (13)		
Less than €800	73.3 (198)	21.2 (42)	0.324	0.613
Grief, %(*n*)				
No	92.59 (250)	19.6 (49)		
Yes	7.4 (20)	30 (6)	1.235	0.258
Cases in the setting, %(*n*)				
No	71.48 (193)	20.2 (39)		
Yes	28.51 (77)	20.8 (16)	0.11	0.916
Psychotherapy, %(*n*)				
No	21.85 (59)	33.9 (20)		
Yes	78.15 (211)	16.6 (35)	7.805	0.005 *
Occupational Therapy, %(*n*)				
No	56.29 (152)	25.7 (39)		
Yes	43.71 (118)	13.6 (16)	6.190	0.013
Nursing, %(*n*)				
No	48.51 (131)	22.1 (29)		
Yes	51.48 (139)	18.7 (26)	0.490	0.484
Loss of employment, %(*n*)				
No	90 (243)	18.1 (44)		
Yes	10 (10)	40.7 (11)	6.589	0.01 *
Symptoms COVID, %(*n*)				
No	85.5 (231)	20.8 (48)		
Yes	14.4 (39)	17.9 (7)	0.169	0.681

N: number of cases. Data expressed as N (%); ^a^ chi-square test; * *p* < 0.05.

**Table 2 healthcare-10-00064-t002:** Bivariate analysis of the number of urgent mental health consultations as a function of different variables.

Variables	Mean ED (SD)	t or F ^a^	*p*
Age	−0.014 (B)		0.409
Gender			
Man	0.82 (2.89)		
Woman	1.2 (3.98)	−0.863	0.389
House hold composition			
Family of origin	1.28 (3.78)		
Own family	0.58 (1.49)		
Single-family	1.45 (5.79)		
Other	0.58 (0.92)	0.990	0.398
Level of education			
Primary	1.35 (4.57)		
Secondary	1 (3.22)		
University students	0.48 (0.88)	1.09	0.338
Diagnosis			
Schizophrenia or other psychotic disorder	1.30 (4.96)		
Personality disorder	1.21 (3.39)		
Bipolar disorder	0.60 (0.78)		
Depressive disorder	0.38 (0.75)		
Other	0.91 (2.87)	0.500	0.736
Economic income			
More than €800	0.46 (1.34)		
Less Than €800	1.24 (4.04)	−2.391	0.018 *
Grief			
No	0.92 (3.3)		
Yes	2.35 (5.76)	−1.09	0.289
Cases in the setting			
No	1.12 (3.99)		
Yes	0.81 (2.24)	0.633	0.527
Psychotherapy			
No	1.30 (3.49)		
Yes	0.95 (3.56)	0.665	0.506
Occupational Therapy			
No	1.26 (4.20)		
Yes	0.72 (2.43)	1.245	0.214
Nursing			
No	0.71 (2.05)		
Yes	1.33 (4.5)	−1.452	0.148
Loss of employment			
No	1.02 (3.69)		
Yes	1.11 (1.62)	−0.120	0.905
COVID Symptoms			
	0.274 (B)		0.002 *

Data expressed as Average (SD); B: Coefficient B established by linear regression; SD: Standard Deviations; ^a^ Student’s *t*-test or ANOVA; * *p* < 0.05.

## Data Availability

The data presented in this study are available on request from the corresponding author. The data are not publicly available because they are part of an ongoing project.

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
