# Peer review of "Risk Factors for Relapse in People with Severe Mental Disorders during the COVID-19 Pandemic: A Multicenter Retrospective Study"

_healthcare, 2021, doi:10.3390/healthcare10010064_

Round 1
Reviewer 1 Report
Respected authors, this is sound retrospective research. I found it to be informative, although lengthy. In my opinion, the introduction section could be reduced by half, as most of the facts are already well known. Tables should be improved as in many instances there are no spaces between words, while used abbreviations are not listed under the table. The discussion section is also sound, although the use of SMD is not only a strength of this research but also a limitation (mixing apples and pears, which I believe is the case with SMD (for example, grouping schizophrenia with personality disorders). Furthermore, if I read correctly, participants were not diagnosed with COVID-19, they were considered infected based on the presence of symptoms associated with covid. If that is the case, it should be stated as a clear limitation (I am aware definitive confirmation was not available at the time, but nevertheless).
Of minor issues, the text should be thoroughly checked as at some instances I found abbreviations such as SMT or SMG (I believe authors meant - SMD)
Reviewer 2 Report
In overall, the paper needs to undergo proofreading before publication, preferably by a native speaker experienced in scientific papers editing.
Abstract:
„SMDs” needs to be explained.
Introduction “
A recent meta-anal-38 ysis provided specific figures for the prevalence of common mental disorders since the 39 virus spread comparable to those observed in the latest study published by WHO” this sentence might be deleted or fixed
“Spanish flu pandemic in the 18th century” are You sure that H1N1 influenza A virus outbreak took place in 18th century? Do you mean pandemic that happed after Ist world war?
MDD” needs to be extended. In overall, all acronyms need to be explained while the first introduction in the text
In overall, the introduction seems to be nice, but it could be shortened and the flow of introduction could be improved. For instance, why two separate paragraphs describe schizophrenia?
Material and methods:
After reading this paragraph I have o idea how the control group looked like? What kind of patients were included as controls? What was difference between two examined groups? What was the serious mental disease group specifics? What kind and prevalence of disorders were there? It could be described in the first paragraph.
„The level of statistical significance for this study is p <0.05..” You should phrase it as „All analyses were performed with a significance level α = 0.05.” or similar
Results
“This section may be divided by subheadings. It should provide a concise and precise 225 description of the experimental results, their interpretation, as well as the experimental 226 conclusions that can be drawn.” Please delete it!
Throughout whole results section You write sentences as “the proportion of admissions was significantly higher”. How it was examined? Please note that the Chi-square test of independence determines whether there is a statistically significant relationship between categorical variables. Therefore, how the differences in proportions were examined? If by Chi square, then I suppose that the wording should be changed throughout the manuscript.
“he mean number of emergency 254 visits was also significantly higher in people with low income (1.24 vs. 0.46; p=0.018). With 255 respect to people who had suffered the loss of a family member due to COVID-19, in our 256 sample they presented a greater number of emergencies, although this was not statisti-257 cally significant (2.35 vs. 0.92; p=0.289).” if that was examined with independent t-test, then I suggest to provide man and SD values and effect size for each comparison.
“Nor were emergencies associated with the socio-demo-260 graphic variables studied (age, sex, household composition or economic level) or with 261 diagnosis.” I would wrote that there was no significant linear relationship between (…) when this was tested using regression analysis.
Why table 2 contains “t or F”? Where it comes from? Why sometimes You use „,” and sometimes „.” In providing decimal places?
Discussion
In overall, the discussion seems to be nice, but it could be shortened and the flow of introduction could be improved, the same as with introduction. Every specific subparagraph of discussion should aim at discussing a specific result from Your result with results from other studies.
“In this multicenter retrospective cohort study based on data collected from 270 pa-267 tients with SMDs who had been followed up in day hospitals during 2020, different factors 268 related to the COVID-19 pandemic were found to be associated with an increased risk of 269 relapse.” I would be more specific here. Please write them down.
“Loss of freedom during lockdown may also lead to secondary losses such as losses 315 of relationships, leisure and social suport” please be more specific here. What kind of freedom was lost? Do You consider government-induced restrictions in social activities?
“The statistically significant association between higher numbers of emergency men-323 tal health visits and COVID-19 symptoms gives us an idea of the fear, worry and anxiety 324 and the general need for emergency hospital care that may be experienced by those suf-325 fering symptoms of this disease, due both to their acute discomfort and to uncertainty 326 regarding the evolution of their illness. Related public health measures such as home iso-327 lation may also lead to a worsening of previous psychopathologies. The WHO has already 328 noted an increased need for care among specific population groups at particular risk of 329 COVID-related psychological distress, including people with a history of mental health 330 issues [52].” This part might serve as an introduction to discussion maybe, after You describe Your results?
- Conclusions
Please be more specific here. What are conclusions from Your own study? After writing them down, You can possibly write bout directions of further research, when You would clearly delineate it from conclusions directly related to You own study.
Round 2
Reviewer 2 Report
Congratulations! The paper has been vastly improved.
I would like to make just two suggestions:
1. The paper might be re-read and editing might be improved. Some sentences are confusing.
2. 2. I would focus on conclusions from Your own study in conclusions section in the main text. In the current version You have described directions of future studies. I would move that part to the last part of discussion under subparagraph "future studies". In the conclusions section I would describe solely conclusion from Your own study.
Author Response
Thank you very much for your interest in our manuscript and for the congratulations for its improvement.
Below, we will answer you about your suggestions for improvement.
Point 1. The paper might be re-read and editing might be improved. Some sentences are confusing.
Response 1: Following your suggestion, the manuscript has been carefully read and changes have been made.
Point 2. I would focus on conclusions from Your own study in conclusions section in the main text. In the current version You have described directions of future studies. I would move that part to the last part of discussion under subparagraph "future studies". In the conclusions section I would describe solely conclusion from Your own study.
Response 2. Thanks for the suggestion. We have made the changes based on your comment.